# Noise Emission Models of Electric Vehicles Considering Speed, Acceleration, and Motion State

**DOI:** 10.3390/ijerph20043531

**Published:** 2023-02-17

**Authors:** Ziqin Lan, Minmin Yuan, Shegang Shao, Feng Li

**Affiliations:** 1National Environmental Protection Engineering and Technology Center for Road Traffic Noise Control, Beijing 100088, China; 2Research Institute of Highway Ministry of Transport, Beijing 100088, China; 3School of Automobile and Transportation Engineering, Guangdong Polytechnic Normal University, Guangzhou 510665, China

**Keywords:** road traffic noise, electric vehicle, noise emission model, acceleration, motion state

## Abstract

Electric vehicles, known for their low-noise emission, are popular and widespread in metropolises in China, and they provide an opportunity for a reduction in environmental noise from vehicles. To understand the noise from electric vehicles better, this study develops noise emission models considering speed, acceleration, and motion state. The model construction is based on the data collected from a pass-by noise measurement experiment in Guangzhou, China. The models describe a linear relationship between the noise level, the logarithm of speed, and the acceleration for multiple motion states (i.e., the constant-speed state, the acceleration state, and the deceleration state). From the spectrum analysis, the low-frequency noise is barely affected by the speed and acceleration, but the noise at a certain frequency is most sensitive to them. Compared to other models, the proposed ones have the highest accuracy and the greatest ability for extrapolation and generalization.

## 1. Introduction

Road traffic noise becomes the main source of environmental noise in large cities due to the densification of road networks and the growing number of vehicles. It has negative impacts on residents’ health, such as cardiovascular diseases, hypertension, cognitive impairments, high sleep disturbance, and annoyance [1,2,3]. The health impacts of road traffic noise raise concern among the policymakers and the general public in China. Since 5 June 2022, the new law on the prevention and control of noise pollution has been put into force by the Chinese government. The law declares that the state must promote the system of standards for the prevention and control of noise pollution. Noise emission models are the foundation of the tools for understanding and reducing traffic noise pollution [4,5,6,7], e.g., dynamic traffic noise simulation [8,9] and traffic noise mapping [10,11,12]. Nowadays, electric vehicles (e.g., electric cars and electric buses) occupy a quite large proportion of vehicles in metropolises on account of incentive policies and the advantage of low-carbon emissions [13] and low-noise emissions [14] in contrast to internal combustion engine vehicles. The characteristics of road traffic noise in metropolises have enormous changes because of the intervention of the increasing number of electric vehicles.

However, the standards, regulations, and guidelines about the noise emission of vehicles have not yet been updated, and some contents do not meet the requirements of assessing road traffic noise in the current environment. The recommended noise emission models in these official documents still maintain the original version, which was derived from the data collected from internal combustion engine vehicles. Thus, to describe the characteristics of noise emission in the new situation and update the relevant content of the traffic noise standards, this study aims to develop noise emission models of electric vehicles by a pass-by noise measurement. The proposed models take into account the speed of vehicles, similar to the existing emission models (e.g., the CoRTN model [15], the RLS 90 model [16], the NMPB model [17,18], the Harmonoise model [19,20], the ASJ-RTN model [21,22]), as well as the acceleration and the motion state of electric vehicles. The contribution of this study is the noise emission models of electric vehicles considering speed, acceleration, and motion state for the traffic noise environment of China.

The rest of the paper is organized as follows. Section 2 reviews the related works of the traffic noise prediction model. Section 3 describes the details of the measurement experiment and the collected data. Section 4 constructs the noise emission models of electric vehicles and conducts some relevant analyses. Section 5 compares the proposed model with other models and tests the performance. Section 6 concludes the paper.

## 2. Literature Review

There are many studies on noise emission models of traditional vehicles. These studies can provide a basic modeling idea for electric vehicle noise emission. We summarize the common noise prediction models, e.g., the CoRTN model, the RLS 90 model, the NMPB model, the Harmonoise noise prediction model, the CNOSSOS-EU model, the FHWA model, and the ASJ-RTN model below:(1)The CoRTN model (i.e., calculation of road traffic noise model) [15] is based on traffic flow through a road segment in an hour, and it also takes into account traffic speed, heavy vehicle percentage, road gradient, pavement type, distance, shielding, angle of view, and reflection.(2)The RLS 90 model (i.e., Richtlinien für den Lärmschutzan a Strassen-90) [16] is a function of the speed limit of roads considering vehicle type, including an additional correction accounting for the increased effect of traffic lights at intersections.(3)The NMPB model (i.e., the French method for road traffic noise prediction) [17,18] consists of a rolling noise component and a power unit noise component, and it takes into consideration traffic flow conditions (i.e., the steady traffic flow and the unsteady traffic flow) and road surface aging.(4)The Harmonoise noise prediction model [19,20], developed in the European projects Harmonoise and Imagine, decomposes the noise emission of a vehicle as the energetic sum of a rolling noise and a propulsion noise, which are both related to the speed of vehicles, and it takes account of the acceleration by using a correction.(5)The CNOSSOS-EU model (i.e., European-common noise assessment methods) [23] includes two parts: a rolling noise and a propulsion noise, which are both correlated to the traffic speed, but it ignores the effect of acceleration and deceleration because it is developed for generating strategic noise maps.(6)The FHWA model (i.e., federal highway administrative model) [24,25] is expressed as a function of vehicle speed, and it uses a term to reflect the traffic flow regime, but it does not represent the acceleration of vehicles explicitly.(7)The ASJ-RTN model (i.e., Acoustical Society of Japan-Road Traffic Noise model) [21,22] describes a linear relationship between the noise level and the logarithm of speed, including some correction terms for road-related factors. It also considers the effect of the traffic flow states (i.e., the steady state and the unsteady state).

Additionally, many scholars improve the above models to achieve better accuracy for traffic noise prediction in some specific situations. Can and Aumond [26] analyzed the sensitivity of estimated road traffic emissions to vehicle kinematics variables and improved the noise emission models through the introduction of additional traffic parameters. Asensio et al. [10] presented a method of calculating the contribution of a specific vehicle to the overall noise pollution, which depended on the type and state of maintenance of the vehicle and road surface, the speed of the vehicle, the acceleration, and the type of driving. Pascale et al. [27] modeled vehicle noise emission considering the impact of kinematic variables. Peng et al. [28,29] developed a six-category heavy vehicle source emission model in free-flowing conditions based on the statewide road setting in New South Wales, Australia. Sánchez-Fernández et al. [30] studied the influence of temperature on the sound pressure level of tire–road noise emissions under actual conditions of continuous vehicle flow. Cai et al. [31] developed a traffic noise emission model on a wet asphalt road based on traffic noise measurements. Abdur-Rouf et al., [32], Gardziejczyk et al., [33], Khajehvand et al., [34], and Yadav et al. [35] developed traffic noise prediction models for urban intersections. These above models both consider traffic speed and traffic flow as main variables and introduce some correction terms that account for the road-related factors and the environmental factors. They belong to linear models, i.e., larger traffic speed and larger traffic flow lead to a higher noise emission level, and they are usually expressed as explicit functions of the vehicle-related factors and road-related factors and are always constructed by using the data collected from statistical pass-by measurement experiments. 

By contrast, some studies use machine learning methods to present the nonlinear relationship between noise emission and its factors [36,37,38]. Whether linear models or machine learning models, they are static models that are suitable for calculating the noise level over a period, but they cannot simulate the dynamic changes in noise emission. To this end, dynamic simulation methods of traffic noise based on microscopic traffic simulation are proposed [8,39,40,41].

However, the above models both account for internal combustion engine vehicles instead of electric vehicles. To our knowledge, the study of noise models of electric vehicles is not as comprehensive as that of internal combustion engine vehicles. Pallas et al. [42] proposed a noise emission model for extending CNOSSOS-EU to light electric vehicles. Lan et al. [14] developed noise emission models for electric cars and electric buses but only took into account vehicle speed. Cesbron et al. [43] studied the potential influence of the road surface on the noise emission of electric vehicles in urban areas by a pass-by noise measurement. Pallas et al. [44] investigated the noise emission of electric trucks at a constant speed and modeled the power unit noise as a function of the engine or motor speed and the rolling noise as a function of the vehicle speed. Czuka et al. [45] compared nine different sets of tires by carrying out pass-by measurements for the modeling of rolling noise from electric vehicles. These models only take into consideration the impact of vehicle speed on the noise emission of electric vehicles. 

In addition to the speed, the acceleration and the motion state also affect the noise emission of electric vehicles. For example, the noise emission of a vehicle in an acceleration state is different from that in a deceleration state, even if the absolute values of the accelerations are the same. Therefore, based on intuitive recognition, this study also takes into consideration the impacts of acceleration and motion state on the noise emission of electric vehicles to provide a better description and improve the estimation accuracy of noise emission.

## 3. Experiment and Data

A field experiment was performed to measure and model the noise emission of electric vehicles and collect pass-by noise levels and motion parameters (i.e., speed and acceleration). The experiment was conducted at Waihuanxi Road, a six-lane road in the Higher Education Mega Center, Guangzhou, China. The experimental situation consists of the flat and dry drainage asphalt pavement, the 21–30 °C air temperature, and the clear surroundings without obstruction. In general, there were few social vehicles running on the road, especially in the morning. The background noise was measured five times, and these sound pressure levels are 43.2, 41.0, 41.5, 42.9, and 42.5 dB(A), respectively. Thus, the effects of non-experimental vehicle noises and background noises were rather negligible.

The measurement method refers to the Limits and Measurement Methods for Noise Emitted by Accelerating Motor Vehicles (GB 1495-2002), China’s official manual of vehicle noise measurement. Before the experiment, a sound level meter (HzAiHua AWA6228+) was calibrated by a calibrator and placed on a side of the measurement road. The microphone of the sound level meter is positioned at 1.2 m above the road surface and 7.5 m away from the center line of the first lane. Two radar speedometers were placed on both sides of the sound level meter symmetrically, and they were ten meters apart. These three instruments were both at the same horizontal plane. The layout of the instruments in the measurement area is shown in Figure 1.

Four electric cars were used in the experiment. They are a kind of sedan produced by Beijing Electric Vehicle Co., Ltd. (Beijing, China), and their models are both BAIC BJEV160. The main parameter configuration of the electric cars is shown in Table 1. It is noted that the noise emission of vehicles is dependent on the size and type of electric vehicles, but the following proposed noise emission models based on this experiment are still applicable to electric cars with a similar parameter configuration.

During the experiment, skilled drivers drove the electric cars one by one through the measurement area in different driving states (i.e., at a constant speed, an accelerated speed, or a decelerated speed). When the car entered and exited the measurement area steadily (i.e., traveled at a constant speed or at a steady acceleration/deceleration), the speed values were recorded by the radar speedometers. These speed values were used to estimate the acceleration of the car. According to the study [46], the acceleration is calculated by Equation (1):(1)$$a=\frac{v_{2}^{2}-v_{1}^{2}}{2S}$$
where S is the distance between the two radar speedometers, i.e., 20 m; $v_{1}$ and $v_{2}$ are the speed values when the car entered and exited the measurement area, respectively. It should be noted that the acceleration was assumed to be constant when the car passed through the measurement area. 

When the car passed by the sound level meter, the A-weighted noise level, and the 1/3 octave band spectrum (from 12.5 Hz to 16 kHz) were collected by the sound level meter, and the instantaneous speed was recorded manually through the car dashboard. In brief, a triple consisting of a noise level, a speed value, and an acceleration value can be obtained after each pass-by measurement.

After the experiment, there are 330 valid triples in total. A total of 117 of them corresponds to the constant-speed state, 101 of them correspond to the deceleration state, and 112 of them correspond to the acceleration state. In this dataset, the range of the noise level is from 51.65 dB(A) to 69.17 dB(A); the speed range is from 24 km/h to 62 km/h, which is in a legal vehicle speed range of urban roads; the acceleration range is from −2.19 m/s^2^ to 2.06 m/s^2^. The detailed statistical result of the grouped data is shown in Table 2.

For a preliminary understanding of the collected data, we used the 1/3 octave band spectra to calculate the noise energy percentage based on the groups mentioned in Table 2. According to the studies [14,46], the noise energy percentage is described as
(2)$$E_{i}=10^{0.1L_{A,i}}$$
(3)$$pe_{i}=\frac{E_{i}}{\sum_{i=1}^{32}E_{i}}$$
where $L_{A,i}$ is the A-weighted sound pressure level corresponding to the *i*-th center frequency in the 1/3 octave band spectrum; $E_{i}$ is the noise energy corresponding to $L_{A,i}$; $pe_{i}$ is the noise energy percentage corresponding to the *i*-th center frequency.

Figure 2 shows the noise energy percentages of the groups based on the motion state and the speed range. Overall, the noise energy is concentrated in the frequency range between 500 Hz and 1600 Hz, and the peak is located at the frequency of 1000 Hz for each group mainly. These are consistent with the results of the work [14]. For each motion state, the noise energy distributes in a wider domain in the low-speed range. As the speed increases, the noise energy converges on a high-frequency range gradually. This indicates that electric vehicles at low speeds are inclined to emit obvious low-frequency noise. At the same speed range, the noise energy of the constant-speed state is more concentrated at the frequency of 1000 Hz than that of the acceleration state or the deceleration state, while the noise energy of the acceleration state at the 500-Hz frequency is higher than those of the constant-speed state and the deceleration state. In contrast with the constant-speed state and the acceleration state, there is an apparent peak at the frequency of 6300 Hz in the deceleration state, which may be caused by the braking and the friction of tires against the road surface. To sum up, different motion states of electric vehicles lead to different noise emissions; thus, it is necessary to construct noise emission models for different states separately.

## 4. Noise Emission Models of Electric Vehicles

### 4.1. Model Construction

In the previous studies [15,16,17,18,19,20,21,22], noise emission of traditional vehicles (i.e., internal combustion engine vehicles) was always expressed as a linear function of vehicle speed. In reality, the noise level of electric vehicles is related to the speed and acceleration of electric vehicles. Specifically, higher speed and higher acceleration lead to larger noise levels. Based on the phenomenon and the previous studies, a linear model is used to represent the relationship between the noise emission of electric vehicles and its relevant factors, i.e., speed and acceleration. The linear model is described as
(4)$$L_{A}=c_{0}+c_{1}∙\mathrm{lg}\left(v\right)+c_{2}∙a$$
where $L_{A}$ is the A-weighted sound pressure level; v is the speed of an electric vehicle; a is the acceleration of an electric vehicle; $c_{0}$, $c_{1}$, and $c_{2}$ are the coefficients determined by least squares methods.

Considering the impact of the motion state of electric vehicles on noise emission, which is mentioned in Section 3, three empirical models are fitted by the collected data for the three motion states, i.e., the constant-speed state, the acceleration state, and the deceleration state. To calibrate the coefficients of the models, we divided the collected data into three parts according to the motion states and then divided each part into a training set and a validation set randomly. The training set consists of 90% of the records of the collected data, and the validation set consists of the remaining records. The proposed models were calibrated by using the training set to obtain the optimal parameters $c_{0}$, $c_{1}$, and $c_{2}$. The final empirical models are as follows:(a)Constant-speed state
(5)$$L_{A}=15.91+28.28\mathrm{lg}\left(v\right)$$

(b)Acceleration state

(6)$$L_{A}=18.52+26.04\mathrm{lg}\left(v\right)+1.10a$$
where a is a positive value.

(c)Deceleration state

(7)$$L_{A}=16.20+31.29\mathrm{lg}\left(v\right)+2.40a$$
where a is a negative value.

The coefficients of determination of the three models are 0.46, 0.54, and 0.67, respectively. Additionally, the *t*-test was used to test the linear significance of these models. The t-test values are both larger than the corresponding critical values, which indicates that there are linear relationships between the noise level of electric vehicles and its relevant factors and proves the rationality of these models. 

### 4.2. Accuracy Analysis of the Models

The performance of the proposed models was tested by using the validation set. In the validation set, there are 11 records belonging to the constant-speed state, 11 records belonging to the acceleration state, and 10 records belonging to the deceleration state. The metric of accuracy is the root mean square error (RMSE), which is described as,
(8)$$RMSE=\sqrt{\frac{\sum_{n=1}^{N}{\left(L_{t,n}-L_{e,n}\right)}^{2}}{N}}$$
where $L_{t,n}$ is the *n*-th truth value; $L_{e,n}$ is the *n*-th estimated value; N is the number of truth values or the estimated values.

The RMSEs corresponding to the constant-speed state, the acceleration state, and the deceleration state are 2.12 dB(A), 1.85 dB(A), and 2.10 dB(A), respectively. These RMSEs are both less than the reference accuracy value, i.e., 3 dB(A), which is suggested by the Good Practice Guide for Strategic Noise Mapping and the Production of Associated Data on Noise Exposure [47]. If the motion state is not taken into consideration, namely that the accuracy verification is conducted on the validation set regardless of the motion state, then the 2.16-dB(A) RMSE is greater than those corresponding to the three motion states. Therefore, the proposed models considering the motion states have a better performance for predicting the noise emission of electric vehicles.

### 4.3. Analysis of the Spectra and the Factors

To understand the impacts of the factors on the spectra, we fitted the linear model between the noise level and the factors for each center frequency. Table 3, Table 4 and Table 5 show the coefficients of determination, the fitting coefficients, and the RMSEs of the fitted linear models corresponding to the center frequencies and the motion states. According to the definition of the coefficient of determination, a larger value indicates a more significant linear relationship between the noise level and the factors. 

In terms of the constant-speed state, the noise is insensitive to the change in vehicle speed in the frequency ranges from 12.5 Hz to 160 Hz and from 12,500 Hz to 16,000 Hz, while the noise is related significantly to the speed of electric vehicles in the frequency range from 400 Hz to 2500 Hz. The maximum coefficient of determination is at the frequency of 1600 Hz and its value is 0.70. The noise level increases with the vehicle speed.

In terms of the acceleration state, the noise is more related to the speed and the acceleration of electric vehicles in the frequency range from 400 Hz to 4000 Hz than in the frequency range from 12.5 Hz to 100 Hz. The maximum coefficient of determination is at the frequency of 1200 Hz, and its value is 0.62. There is a strong and positive relationship between the noise level, the speed, and the acceleration. 

In terms of the deceleration state, the noise is not sensitive to the changes in the speed and acceleration of electric vehicles in the frequency range from 12.5 Hz to 125 Hz, while there are strong linear relationships between the noise level, the speed, and the acceleration in the frequency range from 200 Hz to 4000 Hz. The maximum coefficient of determination is at the frequency of 500 Hz, and its value is 0.82. A larger spend and a larger acceleration lead to a larger noise level.

As shown in Table 6, regardless of the motion states, the coefficients of determination corresponding to the frequency range from 400 Hz to 2500 Hz are greater than 0.3, which indicates that the noise is more related to the speed and the acceleration than in other ranges, but the maximum of the coefficient of determination is at the frequency of 800 Hz, and its value is 0.47. The maximum is smaller than that corresponding to the three motion states mentioned above. Therefore, it is necessary to build the noise emission models of electric vehicles based on the motion states.

## 5. Discussion

### 5.1. Comparison with Other Models

During model construction, we assumed that there was a linear relationship between the noise emission, the speed, and the acceleration of an electric vehicle and used a linear model to fit the collected data. However, some studies use machine learning methods to express the potential nonlinear relationship between noise emission and its factors [36,37,38].

To determine what kind of model is suitable for describing the potential relationship, we compared the proposed models with the other five models on the validation set considering the speed, acceleration, and motion state of electric vehicles. Among the five models, one is a linear model, and the others are often used to capture the nonlinear relationships. The comparative models are as follows:


(1)The first kind of model is a linear model, but it only takes the speed of the electric vehicle into account, i.e., $L_{A}=c_{0}+c_{1}∙\mathrm{lg}\left(v\right)$. This kind of model was common in modeling noise emissions of internal combustion engine vehicles in previous studies. For convenience, this model is named “LR_1para”.



(2)The second kind of model is a support vector regression model (SVR) [48]. Support vector regression is a regression algorithm that supports both linear and nonlinear regressions. There are three preset parameters in the model: a regularization parameter C, an epsilon, and a kernel. In the comparison, the C is 1, the epsilon is 0.2, and the kernel is the radial basis function (RBF) kernel.(3)The third kind of model is a Gaussian process regression model (GPR) [49]. Gaussian process regression also can capture the linear and nonlinear relationship of data. The type of kernel needs to be preset. In the comparison, the chosen kernel is the sum of a dot product kernel and a white kernel.(4)The fourth kind of model is a random forest model (RF) [50]. A random forest is a meta-estimator that fits a number of classifying decision trees on various sub-samples of the dataset. In the comparison, the number of trees in the forest is 100.(5)The fifth kind of model is a multi-layer perceptron regressor (MPR) [51], which is a kind of neural network. In the comparison, the network has a hidden layer with 100 neurons; the activation is the rectified linear unit function ReLu; the learning rate is 0.001; the strength of the L2 regularization term is 0.0001; the solver is Adam, a stochastic gradient-based optimizer.


We trained these models by the training set and calculated the RMSE by the validation set. For the sake of description, the proposed models are named “LR_2para” in the following. Table 7 shows the coefficients of determination and the RMSEs of the six models on four validation sets. The coefficient of determination reflects the goodness of fit of the model and the RMSE represents the predictive ability of the model. Whether in the aspect of the goodness of fit or the predictive ability, the performance of the two linear models (i.e., LR_2para and LR_1para) is superior to that of the other four models (i.e., SVR, GPR, RF, and MPR). The LR_2para model is more suitable for representing the noise emission of electric vehicles than the LR_1para model. This proves that the hypothesis of the proposed models is correct.

### 5.2. Extrapolation Analysis of Different Models

Extrapolation is defined as an estimation of the value of a variable on the basis of its relationship with other variables beyond the original observation range. The ability of extrapolation reflects the generalization ability of models, i.e., the performance of predicting the noise level based on the unobserved values of the correlative factors. To validate the ability of extrapolation of the models, we derived two training sets and two validation sets from the collected data according to the speed range:(1)Situation 1: The validation set consists of the records whose speed values are less than 30 km/h, and the training set consists of the rest of the records.(2)Situation 2: The validation set consists of the records whose speed values are greater than 50 km/h, and the training set consists of the rest of the records.

The six models were trained by the training set and validated by the validation set for Situation 1 and Situation 2, respectively. Table 8 shows the training errors and the prediction errors of the six models in Situation 1. Table 9 shows the training errors and the prediction errors of the six models in Situation 2. With regard to Situation 1, the training error and the prediction error of the proposed model are smaller than those of the other models for each motion state. As for Situation 2, except for the prediction error corresponding to the acceleration state, the performance of the proposed model is superior to that of the other models. These indicate that the proposed model has the best ability of extrapolation in the comparison, and it can well describe the relationship between the noise emission of electric vehicles and its relevant factors.

## 6. Conclusions

To present a better understanding of the traffic noise characteristics of electric vehicles, this study develops the traffic noise emission models of electric vehicles considering speed, acceleration, and motion state by a pass-by noise measurement experiment in Guangzhou, China. The results of the accuracy validation and the comparison between the six models indicate that the proposed models have the highest accuracy and the greatest ability of extrapolation and generalization, and they can provide a better description of the noise emission of electric vehicles. From the spectrum analysis, no matter what state electric vehicles are in, the speed and the acceleration have little effect on the low-frequency noise, but the noise at a certain frequency is most sensitive to the speed and the acceleration.

The limitation of this study is the insufficient consideration for the type of electric vehicles, which also determines the characteristics of noise emission. In future work, we will further investigate the effect of vehicle type on noise emission.

## Figures and Tables

**Figure 1 ijerph-20-03531-f001:**
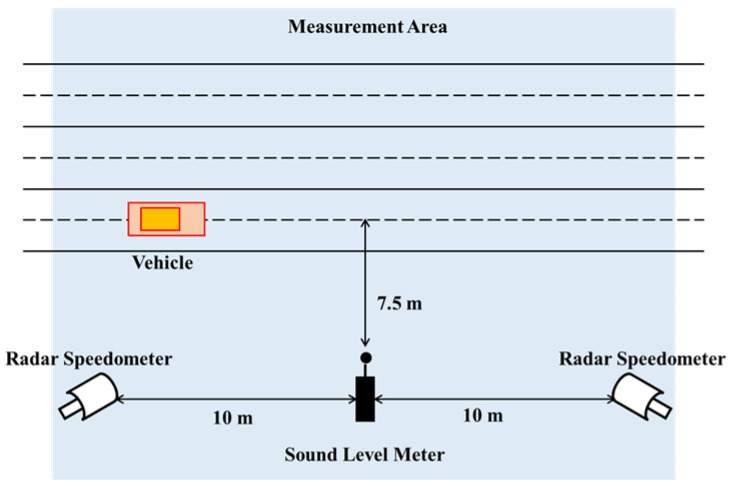
Layout of the instruments.

**Figure 2 ijerph-20-03531-f002:**
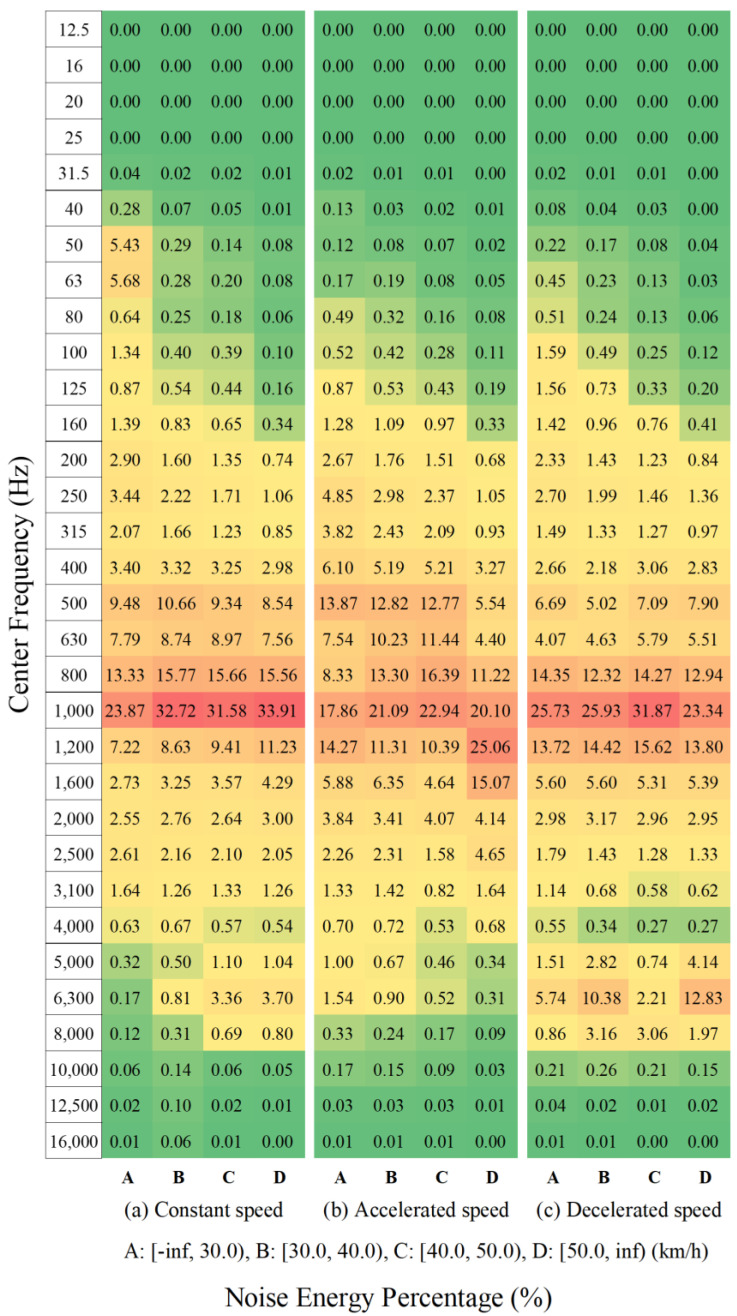
Noise energy percentages of the groups based on the motion state and the speed range (The color of a grid varies from green to red depending on the value of the grid, i.e., a green grid represents a small percentage value and a red grid represents a large percentage value).

**Table 1 ijerph-20-03531-t001:** The parameter configuration of the experimental electric cars.

Configuration	Description
Power type	Pure electric vehicle
Body structure	Hatchback sedan
Engine	45 kW electromotor
Maximum power of motor (kW)	45
Maximum torque of motor (Nm)	144
Energy capacity of battery (kWh)	25.6
Size (mm)	4025 × 1720 × 1503
Wheelbase (mm)	2500
Weight (kg)	1295
Maximum speed (km/h)	125

**Table 2 ijerph-20-03531-t002:** Statistical result of the data grouped by motion state and speed range.

Motion State	Range of Speed (km/h)	Record Number	Range of Acceleration (m/s^2^)	Range of Noise Level (dB(A))
Constant-speed state	<30.0	6	0	[53.21, 58.88]
	[30.0, 40.0)	41	0	[52.72, 62.79]
	[40.0, 50.0)	55	0	[56.50, 66.60]
	≥50.0	15	0	[60.01, 68.71]
Acceleration state	<30.0	14	[0.72, 1.94]	[55.11, 60.21]
	[30.0, 40.0)	57	[0.48, 1.59]	[51.65, 68.39]
	[40.0, 50.0)	35	[0.52, 2.06]	[58.92, 67.41]
	≥50.0	6	[1.32, 2.06]	[62.73, 68.81]
Deceleration state	<30.0	13	[−1.63, −0.69]	[55.60, 60.26]
	[30.0, 40.0)	47	[−1.90, −0.56]	[55.99, 66.23]
	[40.0, 50.0)	32	[−2.19, −0.59]	[61.46, 68.95]
	≥50.0	9	[−2.19, −0.56]	[61.92, 69.17]
Total		330		

**Table 3 ijerph-20-03531-t003:** The coefficients of determination, the fitting coefficients, and the RMSE s of the fitted linear models corresponding to the constant-speed state at different center frequencies.

Center Frequency (Hz)	Coefficient of Determination	RMSE (dB(A))	Center Frequency (Hz)	Coefficient of Determination	RMSE (dB(A))
12.5	0.00	5.64	500	0.33	2.94
16	0.00	6.46	630	0.31	2.41
20	0.01	4.57	800	0.41	3.64
25	0.02	2.80	1000	0.31	3.20
31.5	0.02	2.11	1200	0.64	2.76
40	0.00	2.42	1600	0.70	2.21
50	0.01	3.89	2000	0.52	3.00
63	0.01	2.95	2500	0.32	3.57
80	0.01	1.47	3100	0.21	3.94
100	0.01	2.51	4000	0.22	3.29
125	0.03	1.86	5000	0.27	6.53
160	0.09	1.85	6300	0.16	10.12
200	0.10	2.55	8000	0.11	9.02
250	0.14	2.18	10,000	0.10	5.14
315	0.24	1.61	12,500	0.00	4.54
400	0.41	2.42	16,000	0.00	5.19

**Table 4 ijerph-20-03531-t004:** The coefficients of determination, the fitting coefficients, and the RMSE s of the fitted linear models corresponding to the acceleration state at different center frequencies.

Center Frequency (Hz)	Coefficient of Determination	RMSE (dB(A))	Center Frequency (Hz)	Coefficient of Determination	RMSE (dB(A))
12.5	0.02	4.97	500	0.23	1.60
16	0.02	4.71	630	0.28	2.23
20	0.04	4.50	800	0.37	2.82
25	0.04	2.92	1000	0.53	2.70
31.5	0.06	3.45	1200	0.62	3.23
40	0.00	3.02	1600	0.52	4.05
50	0.05	3.83	2000	0.45	2.84
63	0.05	4.97	2500	0.42	4.35
80	0.00	2.80	3100	0.37	3.80
100	0.04	2.46	4000	0.32	3.52
125	0.13	2.28	5000	0.20	3.49
160	0.17	1.69	6300	0.10	5.13
200	0.15	2.35	8000	0.11	3.98
250	0.09	2.28	10,000	0.05	1.71
315	0.14	1.49	12,500	0.13	2.79
400	0.37	2.01	16,000	0.12	2.65

**Table 5 ijerph-20-03531-t005:** The coefficients of determination, the fitting coefficients, and the RMSE s of the fitted linear models corresponding to the deceleration state at different center frequencies.

Center Frequency (Hz)	Coefficient of Determination	RMSE (dB(A))	Center Frequency (Hz)	Coefficient of Determination	RMSE (dB(A))
12.5	0.04	5.86	500	0.82	1.36
16	0.01	5.03	630	0.73	1.97
20	0.02	4.65	800	0.71	1.86
25	0.04	4.13	1000	0.55	1.85
31.5	0.03	4.06	1200	0.43	1.95
40	0.03	3.76	1600	0.29	2.06
50	0.08	4.55	2000	0.32	2.13
63	0.07	3.53	2500	0.34	2.40
80	0.08	3.43	3100	0.31	2.11
100	0.05	2.02	4000	0.33	1.66
125	0.04	1.89	5000	0.08	8.27
160	0.26	1.71	6300	0.03	12.02
200	0.41	1.58	8000	0.09	10.07
250	0.48	1.77	10,000	0.15	6.73
315	0.60	1.64	12,500	0.10	3.64
400	0.71	2.11	16,000	0.13	3.12

**Table 6 ijerph-20-03531-t006:** The coefficients of determination, the fitting coefficients, and the RMSE s of the fitted linear models at different center frequencies regardless of the motion states.

Center Frequency (Hz)	Coefficient of Determination	RMSE (dB(A))	Center Frequency (Hz)	Coefficient of Determination	RMSE (dB(A))
12.5	0.06	6.10	500	0.38	2.22
16	0.06	5.92	630	0.37	2.40
20	0.04	4.71	800	0.47	2.86
25	0.06	3.10	1000	0.43	2.71
31.5	0.01	3.15	1200	0.44	3.21
40	0.05	2.90	1600	0.36	3.63
50	0.09	3.92	2000	0.33	3.13
63	0.09	3.87	2500	0.30	3.90
80	0.01	2.43	3100	0.25	3.67
100	0.08	2.48	4000	0.25	3.14
125	0.08	2.12	5000	0.10	6.69
160	0.14	1.77	6300	0.04	9.92
200	0.15	2.40	8000	0.06	8.65
250	0.13	2.39	10,000	0.08	4.86
315	0.22	1.81	12,500	0.04	3.50
400	0.43	2.25	16,000	0.02	3.43

**Table 7 ijerph-20-03531-t007:** Comparative results of the six models on four validation sets.

Model		LR_2para	LR_1para	SVR	GPR	RF	MPR
Constant-speed state	Coefficient of determination	0.46	0.46	0.18	0.05	0.63	0.44
	RMSE (dB(A))	2.12	2.12	4.04	4.07	4.52	4.36
Acceleration state	Coefficient of determination	0.54	0.53	0.46	0.13	0.87	0.54
	RMSE (dB(A))	1.85	1.7	3.33	3.06	4.01	3.68
Deceleration state	Coefficient of determination	0.67	0.57	0.08	0.08	0.95	0.65
	RMSE (dB(A))	2.1	2.36	3.16	3.04	4.66	3.59
Regardless of the motion states	Coefficient of determination	0.5	0.47	0.29	0.21	0.81	0.49
	RMSE (dB(A))	2.16	2.26	3.16	3.53	4.52	3.92

**Table 8 ijerph-20-03531-t008:** The training errors and the prediction errors of the six models in Situation 1.

Model	Constant-Speed State		Acceleration State		Deceleration State	
	Training Error	Prediction Error	Training Error	Prediction Error	Training Error	Prediction Error
LR_2para	2.44	2.18	2.12	1.42	1.68	2.09
LR_1para	2.44	2.18	2.12	1.50	1.92	2.72
SVR	3.33	4.87	3.09	3.18	2.42	5.25
GPR	3.20	5.24	2.72	4.14	2.34	5.39
RF	4.13	2.44	3.78	1.61	3.14	2.63
MPR	3.67	2.00	3.39	1.74	2.85	2.26

**Table 9 ijerph-20-03531-t009:** The training errors and the prediction errors of the six models in Situation 2.

Model	Constant-Speed State		Acceleration State		Deceleration State	
	Training Error	Prediction Error	Training Error	Prediction Error	Training Error	Prediction Error
LR_2para	2.40	2.65	2.01	2.49	1.48	4.34
LR_1para	2.40	2.65	2.04	2.25	1.69	4.99
SVR	3.24	4.31	3.15	3.68	2.97	2.47
GPR	3.13	5.06	2.76	4.74	2.84	2.59
RF	3.98	2.41	3.81	3.11	4.00	3.70
MPR	3.60	2.79	3.43	2.36	3.45	3.14

## Data Availability

The data presented in this study are available on request from the corresponding author. The data are not publicly available due to privacy.

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
