# Peer review of "Noise Emission Models of Electric Vehicles Considering Speed, Acceleration, and Motion State"

_ijerph, 2023, doi:10.3390/ijerph20043531_

Round 1

Reviewer 1 Report

This study proposed models for estimating noise emission levels for electric vehicles considering speed, acceleration, and motion state. The data have been collected from an experiment. Overall, the idea is interesting and the paper is well-written. My main concerns are as follows;

- What type of electric car (car make, engine size, whether a sedan or 4X4) was used in the experiment? Please mention these info in Section 3. Noise can be dependent on these variables related to car make as well and in such cases; it is hard to generalize the findings.

- Equation 1 - Apparently, a constant acceleration/ deceleration value has been assumed for the measurement section. Acceleration/ deceleration can be varied over a 20 m stretch. Why a constant value was assumed for acceleration/ deceleration?

- Table 1 - The acceleration and deceleration values are small. And intuitively, noise is higher at larger acceleration/ decelerations.

- Section 4.1 - How these models (eq. 5, 6, and 7) were calibrated? Or in other words, how were the parameters obtained? Were any least square methods or regression methods used? Please provide details.

- Abstract says that "Compared to other models, the proposed model..." (Lines 21-22). Did the authors compare the noise levels obtained from the proposed models with the other existing models?  I cannot see any such comparison or discussion.

- Conclusions mention that “… no matter what state electric vehicles are in, the speed and the acceleration have little effect on the low-frequency noise, …” (Line 369-370). Does this mean that the parameters for speed and acceleration in Eq. 5-7 are not (statistically) significant? Or the output noise levels from equations are statistically the same? As I feel, for such a conclusion, either a formal model calibration (with parameter optimization) should be done OR a sensitivity analysis should be conducted. I cannot see such approaches.

Reviewer 2 Report

See the attached file : Reviewer  report ijerph-2222524.docx

Reviewer 3 Report

In this manuscript, the authors propose a a noise emission model for electric vehicles with the consideration of speed, acceleration and motion state. The manuscript was written in a good structure, but some significant improvements are highly demanded:

1. This field has developed rapidly in recent years, but the latest literature cited in this study is not much. It is suggested that the author can add more introduction about current research in Related Work.

2. The tables occupying too much space in Section 4, but can be summarized in the context, I suggest that the author should put complete tables in the appendix and describe it in detail.

3. The reasons for selecting such comparative models need to be explained in the section 5.1.

4. The paper needs proper proofreading to avoid typos. Please correct the English grammar errors.

Round 2

Reviewer 1 Report

The authors have addressed all of the comments I provided. 

Reviewer 2 Report

All the comments and suggestions are addressed, therefore I recommend the acceptance of the paper.